# Surgical Management after Chiari Decompression Failure: Craniovertebral Junction Revision versus Shunting Strategies

**DOI:** 10.3390/jcm11123334

**Published:** 2022-06-10

**Authors:** Steven Knafo, Mihai Malcoci, Silvia Morar, Fabrice Parker, Nozar Aghakhani

**Affiliations:** 1Department of Neurosurgery, Bicêtre Hospital, AP-HP, University Paris-Saclay, 94270 Le Kremlin-Bicêtre, France; sylvia.morar@aphp.fr (S.M.); fabrice.parker@aphp.fr (F.P.); nozar.aghakhani@aphp.fr (N.A.); 2C-MAVEM National Reference Center for Rare Diseases (Chiari, Syringomyelia and Vertebromedullary Malformations), 94270 Le Kremlin-Bicêtre, France; 3Departement of Neurosurgery, Mondor Hospital, AP-HP, 94000 Créteil, France; mmalcoci90@gmail.com

**Keywords:** Chiari malformation, surgery, foramen magnum decompression, shunt

## Abstract

Revision surgery after posterior fossa decompression for Chiari malformation is not uncommon and poses both strategic and technical challenges. We conducted a single-center retrospective cohort study including all adult patients who underwent revision surgery after posterior fossa decompression for Chiari type I malformation between 2010 and 2019. Among 311 consecutive patients operated on for Chiari malformation at our institution, 35 patients had a least one revision surgery with a mean follow-up of 70.2 months. Mean delay for revision was 28.8 months. First revision surgery was performed at the level of the foramen magnum in 25/35 cases and consisted in duraplasty revision in all cases, arachnolysis (51.4%), additional bone decompression (37.1%), tonsillar coagulation or resection (25.7%), 4th ventricle to cervical subarachnoid spaces shunt (5.7%). Most repeat revisions consisted in CSF diversion procedures, with either ventriculo-peritoneal or syringo-peritoneal shunts. Mean number of interventions per patient was 3.2, with 22.9% of patients undergoing 4 or more surgeries. Based on our experience, we propose that revision at the level of the foramen magnum should be considered as a first-line strategy for Chiari decompression failure. Shunting procedures can be performed in case of extensive arachnoiditis or repeated failures.

## 1. Introduction

Posterior fossa decompression is considered the treatment of choice for symptomatic Chiari malformation type I, with clinical improvement in 80 to 90% and syrinx reduction in 60 to 70% of cases [1,2]. Nonetheless, revision surgery following Chiari decompression is not uncommon, with an estimated reintervention rate of approximately 7% in recent studies [3,4].

Revision surgery following Chiari decompression poses both diagnostic and therapeutic issues. Indications for reintervention range from early postoperative complications, such as cerebrospinal (CSF) leak or pseudomeningocele [3], to delayed arachnoiditis with progressing syringomyelia and neurological deficits [5]. 

Various surgical techniques can be proposed to patients, with two main strategies: either craniovertebral junction (CVJ) revision (i.e., additional bone decompression, duraplasty, tonsillectomy, arachnolysis), or cerebrospinal fluid diversion procedures (ventriculo-, syringo- or subarachnoido- to subarachnoid or peritoneal shunts) [6,7]. 

The goal of the present study was to present a single-center experience in managing these often-challenging cases. We posit that careful pre-revision analysis is key to assess the most common causes for failure and decide whether it is best to perform a revision surgery at the level of the foramen magnum or to proceed to CSF shunting. 

## 2. Materials and Methods

We performed a retrospective cohort study at an adult national reference center for Chiari and Syringomyelia ([BLINDED FOR REVIEW#1]). We screened all patients who underwent posterior fossa decompression for Chiari malformation type I between 2010 and 2019 and included patients who had at least one revision surgery, including those that were initially operated on at other institutions. Ethics committee approval was not required for retrospective observational studies based on medical records under French regulatory texts (Jardé law). Patient consent to use their medical records for research purpose was obtained by the [BLINDED FOR REVIEW#2]). 

Pre- and postoperative clinical and radiological data were extracted from medical records, as well as surgical reports. Only patients with at least 12 months follow-up were included in the analysis. Before revision surgery, all patients had craniovertebral CT-scan and spinal MRI with millimetric T2-weighted and CSF flow sequences. 

Standard surgical procedure for initial posterior fossa decompression performed at our institution included occipital craniectomy with C1 posterior arch ablation, extending beyond the lateral aspect of the spinal cord, opening of the dura, duraplasty with either fascia lata autologous graft or Neuro-Patch^®^ (B-Braun, Saint-Cloud, France). Arachnolysis, tonsillectomy or tonsillar coagulation, fixation or CSF shunts were not routinely performed. Follow-up included a postoperative clinical evaluation 4 to 6 weeks after surgery, a 3-months and 12-months craniovertebral and spinal MRI with CSF flow sequences. 

## 3. Results

Among 311 consecutive patients operated on for Chiari Malformation type I in our department between 2010 and 2019, 35 patients (11.3%) had at least one revision surgery. Mean age was 33.6 (15–55) and sex ratio was F/M = 3/1 (26/9). Among these, 21 patients were initially operated on at our institution (revision rate of 7.1%, 21/297) and 14 patients underwent the initial procedure at another hospital. 

Before the first surgery, most patients presented with impulsive headaches (60%) and pain (65.7%). Syringomyelia was present in most cases (71.4%). All patients had a brain and spinal cord MRI as well as a craniovertebral junction CT-scan. Fundus examination was performed whenever there were clinical or radiological suspicion of idiopathic intracranial hypertension (*n* = 2 patients). No patient had preoperative sign of CVJ instability (e.g., increased AADI or basilar invagination), therefore no dynamic CVJ CT-scan was performed.

Index posterior fossa decompression surgery included C1 posterior arch ablation (82.9%) and duraplasty (65.7%). When performed (*n* = 23), duraplasty was mostly made of either Neuro-Patch^®^ (*n* = 11) or autologous graft (fascia lata: 3 cases; pericranium: 2 cases, material not available for 7 patients, Table 1). Regarding patients without duraplasty, 8 of them underwent occipital craniectomy with ablation of C1 posterior arch, including 1 with dura splitting, and 4 had occipital craniectomy only; none had dural opening without duraplasty.

Mean overall follow-up since the index surgery was 70.2 months (min = 12; max = 310). Revision surgery occurred with a mean delay of 28.8 months (min = 0; max = 264 months). In this case, 14 patients (14/35, 40%) underwent “early” (<12 months) repeat surgery, for either pseudomeningocele (*n* = 8), hydrocephalus (*n* = 4) or persistent papillary edema (*n* = 2), while 21/35 patients (60%) underwent delayed (≥12 months) revision, mostly due to foraminal arachnoiditis (*n* = 18, Figure 1). Patients with early failure (pseudomeningocele, hydrocephalus or persisting papillary edema) had a mean age of 35.3 yo [min = 17–max = 49] and sex ratio F/M = 13/1. Patients with delayed failure (foraminal arachnoiditis) had a mean age of 32.4 yo [min = 15–max = 55] and sex ratio F/M = 13/8.

The most common symptoms leading to revision surgery were progressing sensory or motor deficit (40%), CSF leak and/or pseudomeningocele (28.6%), persisting headaches (17.1%) or neuropathic pain (14.3%). Radiologically, most patients who underwent revision surgery had a progressing (40.0%), newly appeared (11.4%) or stable (22.0%) syringomyelia, while 17% of cases had no syrinx and only 8.6% had a decreased cavity (Table 1). 

Revision surgery was most frequently performed at the level of the foramen magnum (25/35, 71.4%, e.g., Figure 2) and consisted in performing or replacing the duraplasty (in all cases) in association with either foraminal arachnolysis (51.4%); additional bone decompression (37.1%), tonsillectomy or tonsillar coagulation (25.7%), 4th ventricle to cervical subarachnoid spaces shunt (5.7%). Foraminal arachnoiditis was noted in 51.4% of cases (Table 2). 

Non-foraminal CSF diversion was performed in 9 patients (25.7%) undergoing a first revision procedure and consisted in either ventriculo-peritoneal shunt (*n* = 3), a syringo-peritoneal shunt (*n* = 4), lumbo-peritoneal (*n* = 1) or meningocele-peritoneal (*n* = 1; Table 2, Figure 3). 

The mean number of surgical procedures was 3.2 per patient. While most cases underwent only one reintervention (*n* = 21, 60%), 8 patients (22.9%) had 4 or more surgeries (min = 2; max = 16). All patients requiring more than one revision surgery (*n* = 14) eventually underwent a CSF diversion procedure: for these patients, the initial revision strategy was CVJ surgery in 8 cases and shunting procedures in 6 cases. 

## 4. Discussion

In this single-center retrospective study involving 311 patients who underwent posterior fossa decompression for Chiari type I malformation, the revision rate for patients initially operated on at our institution was 7.1% with a mean follow-up beyond 5 years. These findings are in line with previous studies reporting a mean re-operation rate of 6.6% [8] and 6.8% [3]. The leading cause for revisions within the first year was pseudomeningocele, while most delayed revisions were due to progressing syringomyelia. 

Foraminal arachnoiditis was noted in more than half of the patients undergoing revision at the level of the foramen magnum. This observation emphasizes the importance of assessing foraminal scarring with millimetric T2 and CSF flow sequences before proceeding to shunting [9]. Indeed, in our study, more than half (10/18) of the patients who underwent CVJ revision for arachnoiditis did not eventually require a ventriculo- or syringo-peritoneal shunting. It also suggests that foraminal arachnolysis and checking the permeability of the foramen of Magendie during the initial procedure should be performed whenever the dura is opened. 

Before choosing between CVJ revision or shunting, careful radiological analysis should include a cerebral and spinal cord MRI with CSF flow and millimetric T2-weighted sequences to look for foraminal arachnoiditis and assess CSF circulation, but also craniovertebral junction CT-scan to assess the extent of bone decompression. 

If CSF circulation at the craniovertebral junction is still impaired despite the initial decompression surgery, we propose that revision at the level of the foramen magnum should be performed first. This revision surgery should include every possible means to restore a satisfactory CSF circulation upon careful analysis of the diagnostic workup, including additional bone decompression, duraplasty, tonsillectomy or tonsillar coagulation. Whenever foraminal arachnoiditis is encountered, arachnolysis is performed and a shunt between the fourth ventricle and the cervical subarachnoid spaces can be placed if the resulting CSF flow is deemed insufficient. If craniovertebral junction instability has not been addressed initially, fixation should be performed during this first revision surgery. 

When a first revision surgery at the foramen magnum failed to resolve a compressive pseudomeningocele or a progressing syringomyelia, a second revision surgery must be considered after a follow-up period of 6 to 12 months (Figure 4). These failures are most often due to extensive foraminal arachnoiditis and are difficult to treat with a CVJ revision. Hence, we usually perform a CSF diversion procedure: either a syringo-peritoneal shunt if syringomyelia is present, or a ventriculo-peritoneal shunt if it is not. 

These results are limited by the single-center and retrospective nature of our study. Moreover, 14 of the 35 patients included in the analysis were initially operated at other institutions and details regarding the first surgical procedure were obtained from surgical reports. As such, the proposed strategy reflects our current management algorithm rather than guidelines for repeat surgeries after Chiari decompression failures. Multi-center studies including larger numbers of patients and comparing various strategies are required to elaborate a consensus regarding the management of complications following Chiari surgery. 

Existing literature focusing on revision surgery for Chiari malformation is scarce. Klekamp reviewed 61 cases of secondary surgery following decompression for Chiari malformation: of these, 45 procedures were performed at the level of the foramen magnum, including 10 occipito-cervical fusions, achieving clinical stabilization at 5 years in two-thirds of patients [9]. In a systematic review including a total of 616 patients undergoing foramen magnum decompression for Chiari malformation, Schuster et al. found a 6.7% rate of persistent or recurring syringomyelia [4]. For such cases, several revision strategies have been proposed, including syringo-peritoneal or syringo-subarachnoid shunting [10], but only one study has proposed a flow-chart for managing patients with persistent/recurring syringomyelia after Chiari decompression [5]. Altogether, our study is the first to elaborate a comprehensive algorithm for Chiari revision surgery, whether related to progressing syringomyelia or not.

## 5. Conclusions

In conclusion, failure after decompression for Chiari malformation can turn a relatively simple surgical procedure into a very complex situation, with multiple revision surgeries and debilitating neurological consequences. Therefore, it is critical to avoid as much as possible repeated revision procedures, and to acquire it right the first time when a redo surgery is required. Careful clinical and radiological analysis including MRI and flexion-extension CT scans of the craniovertebral junction are compulsory to choose the right revision strategy between CVJ revision or CSF diversion (Figure 4). Multidisciplinary management in reference centers for Chiari and syringomyelia are much-needed whenever first-intention posterior fossa decompression fails.

## Figures and Tables

**Figure 1 jcm-11-03334-f001:**
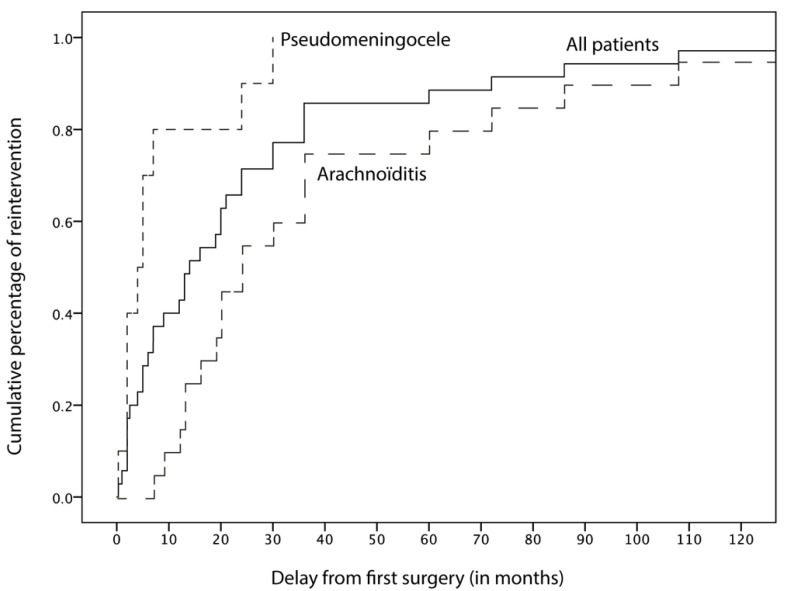
Time to first reintervention following Chiari decompression.

**Figure 2 jcm-11-03334-f002:**
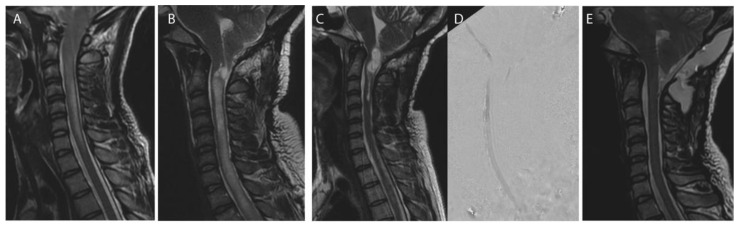
Craniovertebral junction revision after Chiari decompression failure. Case of a 39 years-old female patient initially operated on at another center for a Chiari I malformation without syringomyelia before index decompression surgery (**A**). Foraminal arachnoiditis and progressing syringomyelia (**B**), syringobulbia (**C**) and no CSF flow at the craniovertebral junction (**D**) 7 years postoperatively. CVJ revision was performed: bone decompression was enlarged, duraplasty was removed, and tonsilles were coagulated allowing resolution of the syringomyelia at 1 year postoperative (**E**).

**Figure 3 jcm-11-03334-f003:**
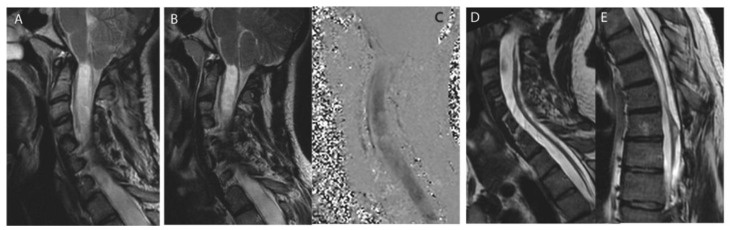
Shunting revision after Chiari decompression failure. Case of a 20 years-olf male patient with a Chiari I malformation with syringomyelia and associated scoliosis before index decompression surgery (**A**). Extensive foraminal arachnoiditis was observed during surgery, confirming the discrepancy between moderate tonsillar herniation and holocord syringomyelia. Persistant (**B**) and circulating (**C**) syrinx with no CSF flow at the craniovertebral junction (**C**). CSF diversion was performed: resolution of the syringomyelia (**D**) after placement of a syringo-peritoneal catheter at the lower extremity of the syrinx (**E**).

**Figure 4 jcm-11-03334-f004:**
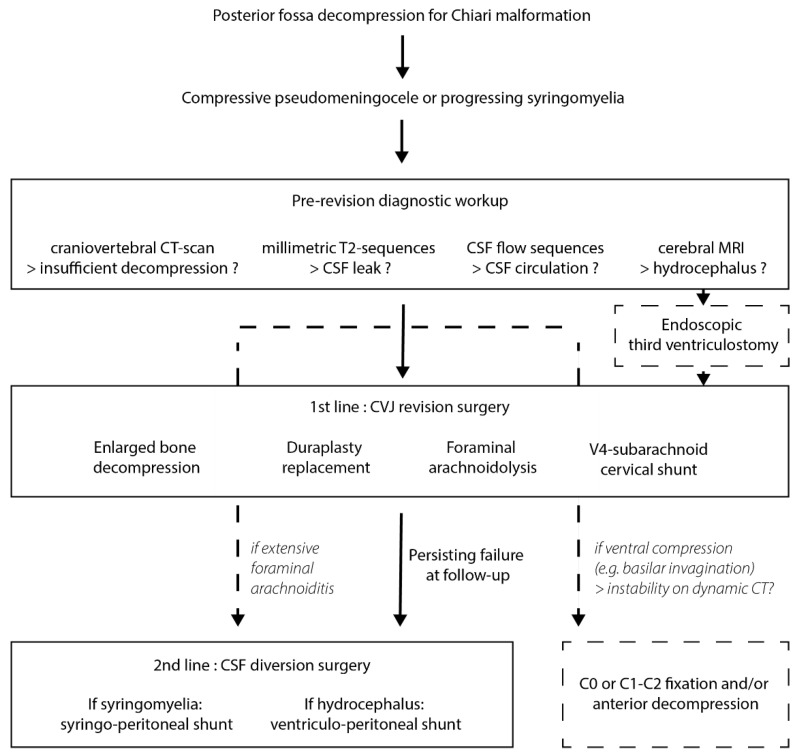
Proposed algorithm for managing Chiari decompression failure.

**Table 1 jcm-11-03334-t001:** Index surgery: presentation and outcome.

		*n* = 35
Initial presentation	Age	33.6 (15–55)
	Sex ratio (F/M)	3/1 (26/9)
	Headaches	21 (60%)
	Sensorimotor deficit	13 (37.1%)
	Neuropathic pain	23 (65.7%)
	Syringomyelia	25 (71.4%)
	Hydrocephalus	2 (5.7%)
Index surgery	Occipital craniectomy	35 (100%)
	C1 ablation	29 (82.9%)
	Duraplasty	23 (65.7%)
Clinical evolution	Headaches	6 (17.1%)
	Neurological deficit	14 (40%)
	CSF leak	10 (28.6%)
	Neuropathic pain	5 (14.3%)
	Papillary edema	2 (5.7%)
Syringomyelia evolution	No syrinx	6 (17.1%)
	Appeared	4 (11.4%)
	Decreased	3 (8.6%)
	Stable	8 (22.9%)
	Increased	14 (40.0%)

**Table 2 jcm-11-03334-t002:** Revision surgery: indication and technique.

Delay from index surgery	Mean	28.8 (0–264)
	Early (<12 months)	14 (40%)
	Late (≥12 months)	21 (60%)
Indication	Arachnoiditis	18 (51.4%)
	Pseudomeningocele	10 (28.6%)
	Hydrocephalus	5 (14.3%)
	Insufficient bone decompression	5 (14.3%)
Technique	CVJ revision	25 (71.4%)
	Decompression	13
	Arachnolysis	18
	Tonsillectomy	9
	Duraplasty	25
	V4-SA shunt	2
	CSF diversion	9 (25.7%)
	Ventriculo-peritoneal	3
	Syringo-peritoneal	4
	Others *	2
Mean number of interventions per patient		3.2 (2–16)

* lumbo-peritoneal (*n* = 1); meningocele-peritoneal (*n* = 1).

## Data Availability

Data are available on request to the corresponding author.

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
