# Peer review of "Surgical Management after Chiari Decompression Failure: Craniovertebral Junction Revision versus Shunting Strategies"

_jcm, 2022, doi:10.3390/jcm11123334_

Round 1

Reviewer 1 Report

I read with great interest this article on the revision surgery of patients with Chiari 1 and syringomyelia. I have a similar experience in the management of these patients and I generally agree with most of the algorithms of treatment reported by the authors. 

Nonetheless, I think the paper should be improved with some adjuncts.

1) Age and sex of the patients are not mentioned. I find this info crucial in this kind of study. Is there any pediatric or ex-pediatric patient in their series?

2) They reported that 23/35 patients received duroplasty. However, they only mention 11 with Neuropatch and 5 with autologous in the paper. What about the other 7 patients?

3) 12/35 did not receive a duroplasty. What operation did they receive? Osteo-ligament foramen magnum decompression only? Durotomy with arachnoid sparing? Durotomy with arachnoid opening +/- other procedures? 

4) What is the age and sex of the patients according to their complications?

5) I think Table 1 and Table 2 should be improved with the above information.

6) It would be useful if they mentioned their standard pre-operative workout in patients with Chiari in the methods before the first operation. For instance, how do they make sure the Chiari is not secondary to IIH? (Do they check the fundus in all their patients preoperatively as part of the neurological examination?). Do they do any preop investigation to exclude CVJ instability? etc...

7) How many surgeons performed the procedures, if known?

8) In the final algorithm, in my experience, I would say that if hydrocephalus: ETV should be attempted before placing a VP shunt; and if refractory syringomyelia: a syringo-peritoneal shunt should not be the first option, but rather the last one. Syringo-subarachnoid shunt (SSS) should be the first option as well demonstrated by the publications in references 8 and 9 (Soleman J et al.) rightly cited by the authors.

Author Response

1) Age and sex of the patients are not mentioned. I find this info crucial in this kind of study. Is there any pediatric or ex-pediatric patient in their series?

Mean age as well as min and max are mentioned in Table 1, while sex ratio was indeed missing. We added age and sex ratio both in Table 1 and in the manuscript (line 70).

2) They reported that 23/35 patients received duroplasty. However, they only mention 11 with Neuropatch and 5 with autologous in the paper. What about the other 7 patients?

For 7 patients, a duraplasty was mentioned in the surgical report although the material wasn’t described. We added this number in the text (line 83).

3) 12/35 did not receive a duroplasty. What operation did they receive? Osteo-ligament foramen magnum decompression only? Durotomy with arachnoid sparing? Durotomy with arachnoid opening +/- other procedures? 

Regarding patients without duraplasty, 8 of them underwent occipital craniectomy with ablation of C1 posterior arch, including 1 with dura splitting, and 4 had occipital craniectomy only; none had dural opening without duraplasty (added lines 83 to 87)

4) What is the age and sex of the patients according to their complications?

Patients with early failure (pseudomeningocele, hydrocephalus or persisting papillary edema) had a mean age of 35.3 yo [min=17- max=49] and sex ratio F/M = 13/1. Patients with delayed failure (foraminal arachnoiditis) had a mean age of 32.4 yo [min=15-max=55] and sex ratio F/M=13/8 (added lines 103-106).

5) I think Table 1 and Table 2 should be improved with the above information.

Age and sex ratio have been added to Table 1

6) It would be useful if they mentioned their standard pre-operative workout in patients with Chiari in the methods before the first operation. For instance, how do they make sure the Chiari is not secondary to IIH? (Do they check the fundus in all their patients preoperatively as part of the neurological examination?). Do they do any preop investigation to exclude CVJ instability? etc...

All patients had a brain and spina cord MRI as well as a craniovertebral junction CT-scan. Fundus examination was performed whenever there were clinical or radiological suspicion of idiopathic intracranial hypertension (n = 2 patients). No patient had preoperative sign of CVJ instability (e.g., increased AADI or basilar invagination), therefore no dynamic CVJ CT-scan was performed (added lines 74-79)

7) How many surgeons performed the procedures, if known?

Surgeries were performed by 6 surgeons from our neurosurgical department (in addition to the 14 patients operated initially by different surgeons from other institutions)

8) In the final algorithm, in my experience, I would say that if hydrocephalus: ETV should be attempted before placing a VP shunt; and if refractory syringomyelia: a syringo-peritoneal shunt should not be the first option, but rather the last one. Syringo-subarachnoid shunt (SSS) should be the first option as well demonstrated by the publications in references 8 and 9 (Soleman J et al.) rightly cited by the authors.

We fully agree with the reviewer and modified the Figure 4 accordingly.

Reviewer 2 Report

Nicely presented manuscript, centered on treatment options available after failure of decompression for Chiari I patients. Although there are inherent limitations, which are stated in the manuscript, it remains a valuable report, describing a single center experience of such cases.

You have mentioned that arachnoidolysis was not routinely performed as a standard treatment option at the initial operation of these patients. Albeit it is not a universally accepted practice, it seems that it offers significant advantages in the overall outcome. The fact that foraminal arachnolysis was a commonly used surgical option during reoperations is propably consistent with that concept. It could be assumed that if arachnoidolysis was performed at the first operation, probably the percentage of Chiari decompression failure could be reduced.

Author Response

You have mentioned that arachnoidolysis was not routinely performed as a standard treatment option at the initial operation of these patients. Albeit it is not a universally accepted practice, it seems that it offers significant advantages in the overall outcome. The fact that foraminal arachnolysis was a commonly used surgical option during reoperations is propably consistent with that concept. It could be assumed that if arachnoidolysis was performed at the first operation, probably the percentage of Chiari decompression failure could be reduced.

Indeed, foraminal arachnoidolysis was not always performed during initial surgery. In our department, we now always ensure the permeability of the foramen of Magendie whenever we perform a duraplasty, i.e. whenever there is a syringomyelia. However, the current retrospective study cannot answer whether this practice would result in a lower revision rate (added lines 185-187).

Round 2

Reviewer 1 Report

I think the authors have satisfactorily replied to all queries raised in the first review and the clarity of the article has greatly improved. There are some minor english misspells (i.e.: spina cord etc..) to correct.

Limitations of the study are approprately stated and conclusions are sound.

Author Response

Thank you for your positive feedback. We have checked again the manuscript for spelling errors (e.g. "spina cord") and language harmonization (e.g. arachnolysis in all occurence rather than arachnoidolysis). 
